# EFFICIENT SELF-REVIEW FRAMEWORK FOR ENHANCING INSTRUCTION FOLLOWING CAPABILITY OF LLM

## ABSTRACT

Large language models (LLMs) often struggle with faithfully following complex instructions, particularly when adherence requires satisfying multiple structural and content-level constraints. While recent work has explored iterative revision or external evaluation to address these limitations, such approaches either incur high computational costs or degrade generation quality through excessive revisions. We introduce Re5, a self-evaluation and revision framework that enables LLMs to improve instruction adherence in a scalable and quality-preserving manner. Re5 decomposes instructions into task and constraint components, performs structure-aware evaluations to prevent error accumulation, and applies selective, constraint-specific revisions. Crucially, this process yields refined outputs that can be fed back into alignment tuning, establishing a data-centric loop for continual improvement. Experiments show that Re5 achieves instruction-following performance comparable to models trained on data from closed-source generators such as GPT-4o-mini, while requiring significantly less data and maintaining high response quality (64.24% win rate over unrevised responses). These results highlight Re5 as a step toward self-improving LLMs, demonstrating how fine-grained self-evaluation can serve as an effective signal for alignment without heavy reliance on external supervision.

## 1 INTRODUCTION

With the advancement of large language models (LLMs), it has become possible to accurately perform a wide range of tasks requested by users. However, adhering user-specified output formats and instructions remains an area in need of improvement. Numerous studies have shown that LLMs often reflect user instructions only partially or overlook formal requirements Harada et al. (2025); Mu et al. (2024); Zhou et al. (2023b); Lu et al. (2023); Sun et al. (2023); Yao et al. (2024); Jiang et al. (2024). To address these limitations, various approaches have been proposed to enhance instruction-following capabilities.

Among these approaches, alignment tuning using high-quality instruction-following data generated by powerful models (e.g., GPT series) has emerged as an effective strategy. However, several studies have shown that large language models (LLMs), including high-capacity models, struggle to fully adhere to complex instructions in a single generation.

To address this limitation, non-training-based methods applicable across different LLMs have been proposed. Among them, iterative revision loops an approach in which the model's initial response is evaluated and revised based on collected feedback have gained attention as a means to improve instruction-following capabilities Pan et al. (2024). However, this method inherently involves repeated generations, and increased reliance on high-performance models leads directly to substantial cost concerns.

As a cost-effective alternative, recent research has explored leveraging open-source LLMs. To compensate for the insufficient self-revision capabilities of these models, external evaluation mechanisms such as high-performance tools or human annotators have been introduced to provide accurate and consistent feedback during the revision process.

Nonetheless, excessive revision aimed at maximizing instruction-following capability can result in a decline in overall response quality. This degradation stems not only from the limitations of the

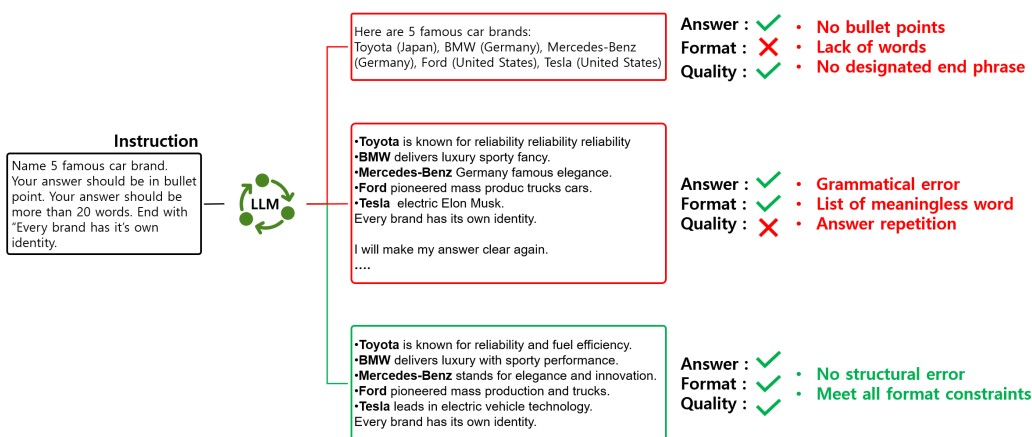

Figure 1: Example of problems with not considering instruction following and with only aiming for instruction following

LLMs themselves but also from the inadequacy of current evaluation methods, which often rely solely on benchmark-based instruction compliance metrics.

As a result, issues such as redundancy, incoherent word sequences, and grammatical errors may go undetected, ultimately compromising the quality of the generated output. Some studies report a win rate of less than 50% when comparing revised responses to the initial outputs, or omit such quality comparisons, thereby overlooking the limitations of self-review-based approaches that focus solely on improving instruction-following performanceFerraz et al. (2024); Zhang et al. (2025).

This suggests the importance of balancing the goal of improving instruction adherence with the need to maintain overall response quality. Evaluation and revision strategies should not only prioritize instruction-following but also incorporate multifaceted quality criteria such as consistency, grammatical, and informativeness.

To address the challenges of instruction adherence, response quality, and efficiency, this study proposes a novel framework based on open-source LLMs that mitigates generation quality degradation in iterative revision settings while achieving a balanced improvement in instruction-following capability. The proposed framework consists of the following key components: First, the user's input is analyzed according to predefined task types and constraints (e.g., format, length, inclusion/exclusion of specific terms), and appropriate evaluation criteria are automatically configured for each constraint. The generated response then undergoes structural evaluation, where grammatically incorrect or structurally unnatural outputs are flagged for regeneration. Subsequently, content-based evaluation is performed using the predefined criteria, and feedback for each category is provided to prompt the LLM to revise and improve its response iteratively. This process enables the enhancement of not only instruction adherence but also structural completeness and content quality. Through benchmark evaluations, our method demonstrated instruction-following performance comparable to that of high-performing models, even when using open LLMs and a small amount of data. Additionally, response quality comparisons using LLM-as-a-Judge showed a win rate exceeding 64% over initial generations.

The main contributions of this study are as follows:

1. Efficiency through self-review : We systematically defined and categorized various types of constraints, and subsequently designed specialized self-evaluation methods tailored to each category. This approach addresses the reliability issues commonly found in prior self-evaluation research, enhancing efficiency by enabling open-source LLMs to achieve performance comparable to resource-intensive high-performance models even with a small amount of data.

2. Avoiding Response Quality Degradation : By assessing structural aspects such as grammatical errors and irregularities in response formats, we identified issues in generation

and guided regeneration. This helped maintain answer quality across iterative generation loops. Moreover, by inducing standardized output formats, we extracted only high-quality feedback from the overall feedback and supplied it to the model, preventing performance degradation due to information overload.

3. Enhancing Instruction Following Capability : We developed detailed evaluation methods for each constraint, including precise evaluation criteria, illustrative examples, and schemes to accommodate LLMs' weaknesses (e.g., in character counting). These allowed the model to clearly interpret each constraint and generate accurate feedback, significantly improving instruction adherence. To prevent instruction-following performance from compromising the accuracy of the user's core task, we also incorporated content evaluation for the task itself.

## 2 EVALUATION PREPARATION

We pre-designed techniques to ensure accurate evaluation within a self-evaluation-based framework. All processes were designed by verifying formats recognizable by the model through in-context learning. First, commonly used instruction-following constraints were categorized, and appropriate self-evaluation methods were devised for each constraint type. For efficient design, constraints with similar characteristics and evaluation methods were grouped together. Additionally, an extraction format was developed to provide high-quality feedback.

### 2.1 CONSTRAINT CATEGORIZATION

To enable precise evaluation of generated output quality, this study categorizes user requests by task type and constraints. Based on an analysis of publicly available instruction-following benchmark datasets and prior research, we classify constraints into two primary categories and four subcategories according to their characteristics.

The first primary category, **Task**, refers to the type of user-desired operation, such as question answering (QA), summarization, and other goal-oriented tasks. Another primary category, **Constraint**, denotes the structural and content-related conditions that must be satisfied in the generated response. We further divide constraints into four subcategories: **Format**, which concerns structural formatting requirements such as bullet points, paragraphs, or organizational patterns like introduction–body–conclusion; **Numeric**, which specifies quantitative requirements such as the number of items, keywords, or listed elements; **Length**, which constrains the size of the response globally or for specific sections, defined by minimum or maximum character or word counts; and **Content**, which imposes semantic restrictions, including mandatory inclusion of specific keywords, exclusion of prohibited terms, or requirements to end with a particular sentence.

### 2.2 EVALUATION METHOD CONSTRUCTION

The goal of this study is to construct an efficient framework that minimizes reliance on external tools by leveraging the self-evaluation capabilities of LLMs to ensure both accuracy and efficiency. To achieve this, we developed detailed self-evaluation prompts tailored to each constraint and task.

All constraints include specific scoring items, criteria for sub-scores within each item, and illustrative evaluation examples. For the Length constraint, which involves calculating the number of characters in a specific part of the generated output, we identified through preliminary experiments and prior studies that LLMs tend to be inaccurate when the required character count becomes large. To address this limitation, we employed external tools only for the Length constraint to compute the actual character count of the generated text. Self-evaluation was then performed by checking whether this count fell within the specified range of the Length constraint. This method transformed the task from exact character counting into a range verification task, thereby enhancing evaluation accuracy.

Although the Numeric constraint also involves numerical reasoning, it typically requires counting various elements, making it inefficient to prepare multiple external tools for different types of counts. Moreover, since the numbers involved are usually small and the task focuses on counting specific

elements, we found it more effective to have the LLM identify and count these elements directly. Therefore, the evaluation for the Numeric constraint is conducted without external tool assistance.

## 2.3 STRUCTURED FEEDBACK EXTRACTION

Previous studies have shown that excessive input length can degrade LLM performance due to information loss or reinforcement of unintended biasesLiu et al. (2024); Anonymous (2025); Hsieh et al. (2024); Levy et al. (2024). Even state-of-the-art models like GPT struggle to maintain full control over generation when exposed to overly verbose or noisy inputs. Therefore, to ensure high-quality responses, it is essential to provide only refined and concise information during the revision process. To this end, we defined a precise output format for LLM-based self-evaluation that includes a concise summarized feedback encapsulating the overall evaluation result and provided it to the model. This structured approach enabled us to extract reliable and standardized feedback, which was then used to guide further revisions.

While this strategy led to a higher number of evaluation failures due to format mismatches in some cases, we considered this a reasonable trade-off. Without high-quality, structured feedback, it becomes difficult to produce high-quality generations. Thus, prioritizing structured precision over extraction recall proved beneficial for overall performance.

## 3 RE5 : RESPONSE RECONSTRUCTION AND REGENERATION FOR REFINED REPRESENTATION

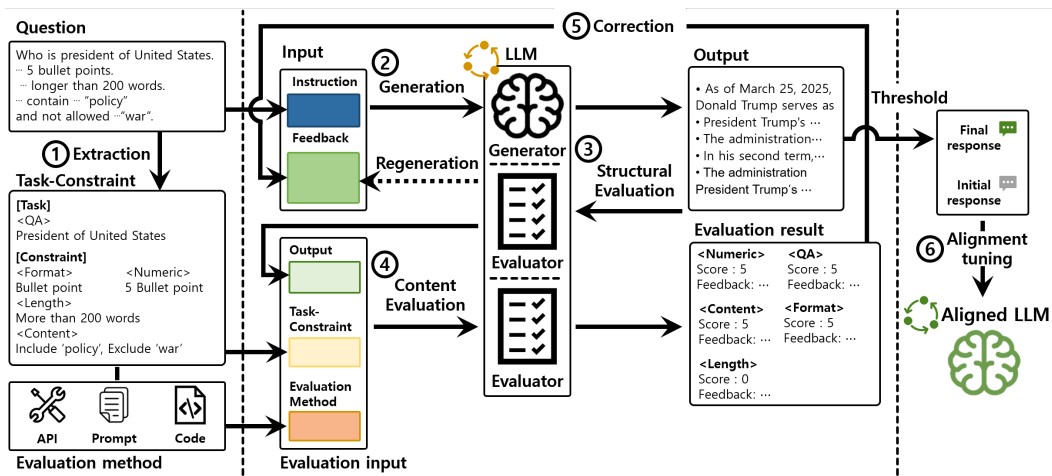

Figure 2: Overall flow of Re5

In this study, we suggest Response Reconstruction and Regeneration for Refined Representation(Re5) framework, a structured pipeline for improving LLM instruction-following performance through an iterative process while maintaining the structural quality. It begins with the extraction of task and constraint elements from user instructions to enable consistent evaluation and feedback. The generation phase produces an initial response, which is refined through multiple feedback-informed iterations. A structural evaluation step ensures that output is well-organized and avoids issues like redundancy before assessing the content. Content evaluation then scores the response against each constraint type separately to allow precise feedback. Lastly, alignment tuning uses high-quality outputs to guide models for instruction following.

## 3.1 EXTRACTION

To ensure accurate instruction following, the first stage of the proposed framework involves the precise extraction of core elements Task and Constraints from the user-provided instruction. We employ a predefined extraction prompt format that enables automatic structuring of the instruction

into explicit components, such as: Task type (e.g., QA, summarization), Constraint categories (Format, Numeric, Length, Content) and specific conditions associated with each task and constraint This structured information serves as the foundation for both evaluation prompt construction and evaluation strategy definition, thereby ensuring consistency throughout the evaluation and revision process. In this study, a high-performance model was used to extract the above information from instructions to ensure accurate information extraction.

### 3.2 GENERATION

The generation process begins with a simple prompt that consists solely of the user instruction, which is provided to the LLM to produce an initial response. Following this, the framework is designed to support iterative regeneration loops. In each loop, feedback is generated based on the results of prior evaluations, and the model either generates a new response or modifies the previous one accordingly. This iterative process gradually refines the output, enabling convergence toward high-quality responses that more precisely satisfy the given instruction.

### 3.3 STRUCTURAL EVALUATION

Experimental observations revealed that even when an LLM output complies with the instruction, structural deficiencies such as redundant responses, meaningless phrase repetition, or unclear paragraph segmentation can significantly impair the efficiency of subsequent evaluations and revisions. To prevent this, we applied a structural evaluation process to assess the structural validity of the responses. By evaluating for structural flaws such as those mentioned earlier responses found to contain such issues were excluded from content evaluation. Instead, feedback regarding the structural defects was provided to the model, and the response was regenerated. This approach prevented the accumulation of errors and improved the overall efficiency of the evaluation process.

### 3.4 CONTENT EVALUATION

This stage involves task and constraint specific evaluations based on the instruction elements extracted in Section 3.1. Each constraint is evaluated independently using custom prompts that align with its category (e.g., Format, Numeric, etc.), allowing for fine-grained scoring and targeted feedback. This separated evaluation approach enables focused assessment of individual constraints, thereby preventing interference errors that may arise during joint evaluations. It also allows for the collection of detailed scores and feedback for each constraint. For every evaluation, the model is guided to output a high-quality, concise summary of the overall result in a standardized format. These summaries are then extracted and used to inform subsequent revisions or regeneration requests, providing precise guidance for improvement while avoiding performance degradation due to excessive information.

### 3.5 CORRECTION

While full regeneration based on evaluation feedback is an intuitive approach, it often introduces new errors and causes previously satisfied constraints to be violated. Our experiments confirm that regeneration frequently omits well-formed elements from earlier responses. To mitigate this, the Correction phase avoids full regeneration. Instead, it selectively modifies portions of the existing response based on the evaluation feedback, preserving correctly implemented constraints. This targeted revision strategy improves overall response quality while minimizing regression in previously correct content.

### 3.6 ALIGNMENT TUNING

The ultimate goal of the iterative generation, evaluation, correction loop is to align the model's behavior with user expectations. Final responses that exceed a quality threshold in the evaluation loop can be stored as high-quality samples. These can then be used for reinforcement learning pipelines such as RLAIF (Reinforcement Learning with AI Feedback). In this way, Re5 serves not only as a practical optimization framework for instruction following, but also as a scalable, data-centric alignment loop that contributes to the long-term alignment of LLMs.

# 4 EXPERIMENT SETTING

## 4.1 DATASET

Existing instruction-following datasets primarily serve as evaluation resources and thus tend to suffer from two major limitations: (1) limited data size, and (2) the absence of explicit task definitions within the instruction. To address these shortcomings, we constructed a new dataset by leveraging a diverse set of NLP task-specific datasets. This enables a more comprehensive investigation of how LLMs generate and refine responses under varied constraints. The datasets used in this study span a variety of tasks including question answering, summarization, and structured generation, providing a broad base for evaluating the model's ability to satisfy constraints across domains. Natural QuestionsKwiatkowski et al. (2019) is a large-scale open-domain QA benchmark derived from real Google Search queries. Each data point consists of a natural language question, a set of associated Wikipedia documents, and annotated short and long answers. XL-SumHasan et al. (2021) is a multilingual abstractive summarization dataset comprising over one million document–summary pairs collected from the BBC. Each summary is professionally annotated, ensuring high reliability. Tulu 3Lambert et al. (2024) is an instruction-tuning dataset focused on precise instruction following. It includes diverse, high-quality prompt–response pairs and is specifically designed to enforce fine-grained constraints such as format, style, length, and content. For Natural Questions and XL-Sum, we augmented the original examples which lacked instruction-level constraints by using GPT-4o-mini to generate tailored instructions. This was done by constructing a prompt that included task descriptions, constraint category definitions, and few-shot examples. Each instruction was extended to include 2–5 additional constraints appropriate to the task and query.

A total of 30,000 data samples were used, with 10,000 from each dataset. Among them, only the cases where the final revised response received a higher evaluation score than the initially generated response was classified as successful cases. In this experiment, 11,686 cases were identified successful, and these were exclusively used for DPO training. In contrast, for other experimental settings using the datasets mentioned above, all 30,000 samples were used for DPO training.

## 4.2 METRIC

To evaluate both instruction adherence and response quality, we adopted a two-pronged evaluation strategy. First, to assess instruction-following accuracy, we used benchmark datasets including IFevalZhou et al. (2023a) and Multi-IFHe et al. (2024), which contain diverse instruction–response pairs with varying constraints. We adopted strict accuracy as the evaluation metric: a response is considered correct only if all instructed constraints are satisfied without requiring additional modifications. Second, to determine whether the quality of responses degrades during the constraint satisfaction process, we conducted an Overall Quality Assessment (OQA). This involved comparing responses before and after correction. We used GPT-4o-mini as the judge model for OQA. Two types of assessments were conducted. **OQA-1** evaluates general response quality based on coherence, grammar, and absence of meaningless repetitions. **OQA-2** evaluates both instruction adherence and response quality. Since LLM-as-a-judge evaluations can exhibit order bias, we randomized the comparison order and reported the average score as the final resultPanickssery et al. (2024).

## 4.3 MODELS

Our base model is LLaMA 3.3 70B Instruct. For comparison, we defined five settings.

In the **Zero-shot** setting, the base model applies the Re5 framework's content evaluation without examples or scoring guidelines. It evaluates responses using only the instruction and output, and generates feedback and revised responses without structural evaluation. The **Few-shot** setting is similar to zero-shot, but includes examples in the evaluation prompts; still, no structural evaluation is applied. The **Similar Performance Generated (SPG)** setting involves fine-tuning on chosen responses generated by another model with comparable scale and performance, specifically Qwen-2.5-72B-InstructQwen et al. (2025). The **High Performance Generated (HPG)** setting involves fine-tuning on chosen responses from a stronger model, for which we used GPT-4o-mini. Finally, the **Publicly Available (PA)** setting uses fine-tuning on open instruction-following datasets, specifically the alignment-tuning subset from Tulu 3 designed for post-alignment trainingLambert et al. (2024).

Table 1: Performance comparison in instruction following and Overall quality

| Model | Data | IFeval (acc) | Multi-IF (acc) | OQA-1 (%) | OQA-2 (%) |
|---|---|---|---|---|---|
| Baseline | - | 0.6174 | 0.2496 | - | - |
| PA | 19890 | 0.4621 | 0.1633 | - | - |
| SPG | 30000 | 0.5656 | 0.2497 | 48.71 | 47.38 |
| HPG | 30000 | 0.6432 | 0.3590 | 73.06 | 70.02 |
| Zero-shot | 30000 | 0.6137 | 0.2327 | 50.37 | 51.72 |
| Few-shot | 30000 | 0.6212 | 0.2541 | 43.87 | 46.47 |
| Re5 | 11686 | 0.6433 | 0.3498 | 64.24 | 65.79 |

## 5 RESULTS

In case of Zero-shot, as the process primarily focused on instruction following capability, overall response quality deteriorated, often resulting in issues such as redundant answers and grammatical errors. Consequently, Zero-shot setting showed decreased performance than baseline in every evaluations.

By providing a guide example for instruction following in the Few-shot setting, benchmark results of instruction following performance has slightly increased compare to Zero-shot. However, due to the increased input length, overall quality of generation decreased, causing lower win rate of OQA than Zero-shot setting. Through Zero-shot and Few-shot setting, we can observe that relying solely on a simplistic iterative process of evaluation and refinement absent robust safeguards for accurate assessment or response quality is insufficient for achieving stable improvements in instruction-following performance. Moreover, it is challenging to maintain consistent response quality under such conditions.

Although the PA dataset is designed for instruction following, according to its construction protocol, the "chosen" and "rejected" responses often differ by only one constraint violation. As noted in prior studies, the subtle quality difference between the two makes them suboptimal for training, despite the dataset being of high quality. This led to consistently low scores in instruction-following evaluations.

The SPG setting, which uses models with comparable parameter sizes, failed to demonstrate a significant quality gap between initial generations in OQA. Furthermore, due to the lack of a revision process, a high rate of instruction violations persisted. Additionally, differences in reasoning strategies across heterogeneous models contributed to its lower performance on instruction-following benchmarks compared to baseline models.

In contrast, the HPG setting, showed improved instruction-following performance relative to baseline models without any refinement process due to the high-quality data generate by high-performance model. In response quality comparisons, although LLM-as-a-Judge tends to favor its own outputs, the high win rates over baseline initial generations can be attributed to the superior data quality produced by high-performing modelsPanickssery et al. (2024).

To validate the effectiveness of instruction-following improvement via self-revision, we compared the OQA-1 and OQA-2 results between model groups that employed self-revision and those that did not. The SPG and HPG settings, which did not include revision, showed decreased win rates in OQA-2 where instruction compliance is also evaluated compared to OQA-1, which focuses solely on response quality. In contrast, the self-revised models achieved improved win rates.

Our proposed method, Re5, prevents response degradation through structural evaluation and the extraction of high-quality feedback. By categorizing constraints and applying specialized evaluation methods tailored to each type, Re5 significantly improves instruction adherence. Furthermore, through its self-revision-based pipeline that integrates these techniques organically, Re5 generates high-quality data which follows instruction well, achieving performance comparable to models trained on high-quality outputs generated by high-performing LLMs even with a small amount of data.

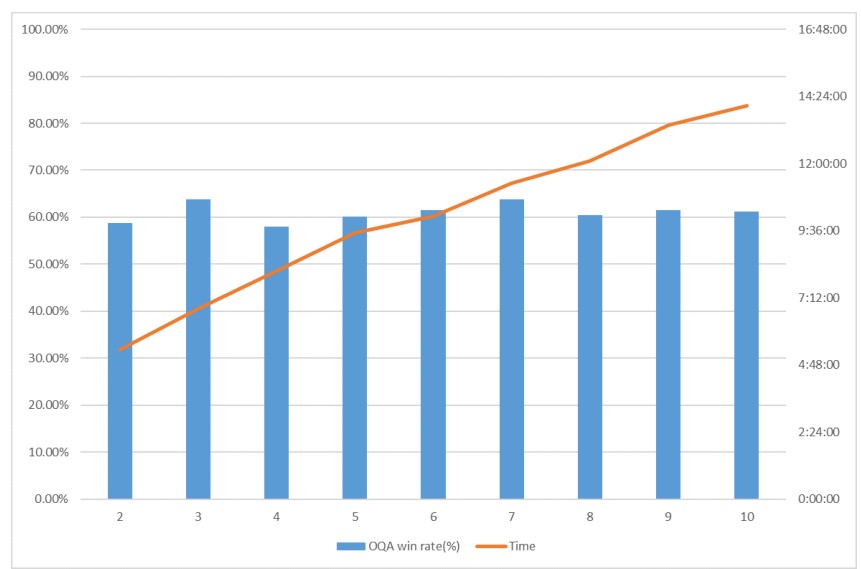

Figure 3: Performance of Re5 with different number of feedback loops

## 5.1 IMPACT OF FEEDBACK LOOP ITERATIONS

Prior studies have reported that iterative self-revision improves response quality up to a certain point, beyond which excessive corrections may lead to diminishing or even negative returns. To empirically validate this phenomenon, we measured both the Overall Quality Assessment (OQA) scores and processing time across varying numbers of feedback loop iterations. As shown in Figure 3, the OQA steadily improved during the initial iterations and saturated after approximately three feedback loops. Beyond this point, additional iterations did not yield noticeable quality gains. However, since each iteration includes several subprocesses such as regeneration, structure evaluation, and content evaluation, the total processing time increased rapidly as the number of loops increased. Based on these findings, we fixed the maximum number of feedback loops to three iterations throughout our experiments to maintain an optimal balance between efficiency and performance.

## 5.2 ABLATION STUDY

The proposed Re5 framework integrates multiple components including structural evaluation, task-specific content evaluation, structured feedback generation, and constraint-wise individual evaluation to generate high-quality feedback and enhance instruction-following performance through effective self-revision. To assess the individual contribution of each component, we conducted a series of ablation experiments under four settings.

In the **w/o Structural Evaluation** setting, structural evaluation prior to content evaluation is removed to test whether early detection and regeneration of structurally flawed responses contributes to overall response quality. The **w/o Task Evaluation** setting disables task-level evaluation and performs only constraint-level assessments, allowing us to examine the impact of explicitly evaluating task fulfillment. The **w/o Structured Feedback** setting assesses whether generating and extracting concise, structured feedback summarizing the evaluation results improves response quality. In this case, the entire evaluation result is provided as feedback to the model without extracting a structured summary. Finally, the **w/o Individual Evaluation** setting examines the effectiveness of applying tailored evaluation methods for each constraint by removing constraint-specific evaluations; instead, a single prompt evaluates all constraints simultaneously, and the model is asked to provide overall scores and feedback.

Each setting was evaluated on two metrics, Evaluation Success Rate (ESR) which rates extraction rate of numerical scores in the self-revision loop and OQA-2 Win Rate

Table 2: Performance comparison according to the presence or absence of each technique

|                             | ESR (%) | OQA-2 (%) |
|-----------------------------|---------|-----------|
| Re5                         | 97.3    | 65.79     |
| - w/o Structure evaluation  | 94.3    | 59.09     |
| - w/o Task evaluation       | 93.3    | 58.16     |
| - w/o Structured feedback   | 89.3    | 51.36     |
| - w/o Individual evaluation | 52.7    | 48.05     |

The absence of structure evaluation was found to cause errors in content evaluation due to structural flaws in the responses. When structure evaluation is applied, structurally invalid responses are regenerated, thereby reducing evaluation errors and improving overall response quality.

Task evaluation ensures that the core task specified in the instruction is preserved throughout the generation process. Without task evaluation, the core content of the task can become distorted during revision, leading to degraded response quality, evaluation errors, and a decrease in evaluation success rates.

Structured feedback prevents generation errors caused by information overload by extracting only the essential elements from otherwise unconstrained model outputs. Generation errors of this kind can propagate to content evaluation, significantly lowering both the evaluation success rate and the overall response quality.

Lastly, individual evaluation not only enables precise assessment by applying methods tailored to each constraint category, but also mitigates interference between constraints during evaluation. Even when all constraints are provided alongside the instruction and the model is explicitly asked to evaluate only a specific constraint, LLMs may still evaluate unintended aspects, leading to unexpected errors. This results in a sharp decline in evaluation success rates and a significant drop in response quality.

# 6 CONCLUSIONS

In this study, we propose Re5, a self-revision-based framework that generates and trains alignment data to improve instruction-following capabilities of LLMs, while addressing limitations in prior work namely, degradation in response quality and inefficiency due to reliance on high-resource, high-performance models. Re5 categorizes instructions based on task types and constraint requirements, applies tailored evaluation methods to precisely assess instruction adherence, detects structural flaws in generated responses, and provides only high-quality feedback to prevent degradation in response quality. By organically integrating these components into a unified self-evaluation and revision framework, Re5 generates high-quality instruction-following output which enables open LLMs to achieve instruction-following performance comparable to that of models trained on high-quality data from high-performance LLMs even with a small amount of data, offering a more resource-efficient alternative.

Looking forward, we plan to extend this framework beyond the LLaMa 3.3 70B model, exploring its applicability to smaller-scale models and diverse architectures such as Qwen, to better understand its generalizability. In addition, while current benchmarks show Re5 reaching performance on par with strong baselines, future work will focus on further enhancing both benchmark metrics and the qualitative generation quality of responses. Taken together, this work highlights that even with a small amount of self-generated data, Re5 can achieve performance comparable to models trained with large-scale, high-quality external data, paving the way for broader and more efficient applications.

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

APPENDIX

# A  RELATED WORKS

The iterative self-revision process involves repeating the same revision steps to enhance the quality of initially suboptimal outputs from LLMs, often by modifying responses or extracting additional information. Below, we describe various studies that have addressed poor performance in specific domains and tasks or have met complex custom format requirements.

## A.1  ENHANCING TASK PERFORMANCE

ReSearchPiché et al. (2024) improves response accuracy by decomposing an answer into fine-grained information units and evaluating the factuality of each. Correct information is retained and injected into the regeneration prompt, allowing the model to correct inaccuracies while preserving valid content.

Self-RefineMadaan et al. (2023) is a feedback-based framework that enhances performance on tasks such as code generation, mathematical reasoning, and acronym creation using high-capacity LLMs like GPT-3.5 and GPT-4. It provides concrete, actionable feedback and few-shot examples to guide the model toward optimized answers.

ReflexionShinn et al. (2023) introduces a reinforcement learning framework incorporating three LLM-based modules: Actor, Evaluator, and Self-Reflection. The Actor generates actions, the Evaluator assesses results and provides rewards, and the Self-Reflection module analyzes failure cases, generating reflective feedback stored in memory. The Actor improves by incorporating these reflections, iteratively enhancing performance over time.

## A.2  ENHANCING INSTRUCTION COMPLIANCE

Decompose, Critique, and Refine (DeCRIM)Ferraz et al. (2024) introduces a self-correction pipeline that separates, evaluates, and refines model outputs to increase adherence to user instructions. By decomposing requirements and using high-performance tools for evaluation, DeCRIM enhances compliance. However, evaluations using LLM-as-a-Judge for Overall Quality Assessment (OQA) revealed a low win rate, indicating that excessive focus on instruction adherence may compromise overall response quality. Divide-Verify-Refine (DVR)Zhang et al. (2025) aims to improve instruction compliance by decomposing user requirements, evaluating outputs with external tools, and reusing evaluation results. The model first uses prompting techniques to identify requirements and assign them to appropriate categories. Then, it employs suitable external tools for evaluation and stores results as reusable few-shot examples for similar future tasks. However, reliance on LLMs to select external tools can lead to poor evaluation outcomes due to inappropriate tool selection.

# B  PARAMETER

We use vLLM to load model and send request for the generation and evaluation. Below is the parameter value of request and values used in our framework. Evaluation threshold is the score from the evaluation and is the basis for the score to be provided as feedback.

- Generation
    - Temperature : 0.7
    - Frequency penalty : 0.8
    - Repetition penalty : 1.2
    - Max tokens : 500
- Structural evaluation
    - Temperature : 0.0
    - Frequency penalty : 0.8
    - Repetition penalty : 1.2

- Max tokens : 200

• Content evaluation

- Temperature : 0.0
- Frequency penalty : 0.8
- Repetition penalty : 1.2
- Max tokens : 500

• Threshold

- Evaluation : 4
- Score : 100

# C PROMPTS

## C.1 EXTRACTION

We use few-shot prompt with 10 examples of diverse task and constraint. Each task and constraint can be added, removed, and modified to suit your needs. [Caution] specifies the areas where LLM is vulnerable or confused, which can help in extracting information.

---

**Task-Constraint Extraction**

You are an AI assistant.
Your job is to analyze the given [Instruction] and accurately classify the user's request into two categories: [Task] and [Constraint].

[Task]: The primary objective that the user wants the model to perform. This refers to core NLP tasks such as Question Answering (QA), Summarization and Generation.
[Constraint]: The user's specific requirements that define how the model's output should be structured. These constraints modify or refine the response format. Constraints include: Length, Format, Numeric, Content

[Task]
<Question Answering (QA)>: This task involves understanding a given context and accurately extracting or generating an answer in response to a question. The key objective is to comprehend the meaning of the input text, identify relevant information, and provide a precise and contextually appropriate response. QA can be fact-based (extracting exact information) or generative (producing well-formed answers).
<Summarization>: This task requires condensing a given text while retaining its core meaning and essential information. The goal is to produce a shorter version that captures the main points in a coherent and concise manner. Summarization can be extractive (selecting key sentences) or abstractive (rephrasing content in a new way).
<Generation>: This task involves producing new text based on a given prompt or input. The objective is to create coherent, contextually appropriate, and meaningful content that aligns with the user's request. Generation can vary in complexity, ranging from composing a detailed response to an open-ended question to transforming or expanding a given text while maintaining relevance and fluency.

[Constraint]
<Length>: This constraint controls the length of the response. It can be specified in terms of word count (e.g., 'no more than 50 words' or "400 word at least") or described qualitatively (e.g., 'keep it short' or 'provide a detailed response').
<Format>: This constraint defines the structural format of the response. The response can be formatted as bullet points, a structured essay with an introduction-body-conclusion, a numbered list, or any other specified layout.
<Numeric>: This constraint determines the number of target i.e. specific constraint or information units included in the response. It can specify an exact number of items (e.g., 'list three key factors') or set a range (e.g., 'provide 5-10 examples') with an expression

---

such as "less than", "at least", "more than", "between"

<Content> : This constraint specifies mandatory content that must be included or excluded in the response. It is divided into two subcategories:

(Include): This subcategory specifies mandatory content that must be included in the response. Such as begin with or conclude with a specific phrase or sentence. Or a specific phrase or sentence must appear somewhere in the response.

(Exclude): This subcategory defines content that must be excluded from the response. Such as not begin with or conclude wihth a specific phrase or sentence. Or a specific phrase or sentence must not appear anywhere in the response.

Example 1
[Instruction]
Please tell me at least three things to do for healthy muscle growth. Please write as a bullet point of 150 characters or less and do not start with "Yes, I understand" but end with "Thank you.". Make sure to use the word protein at least 8 times, and the word muscle appears less than 10 times.

[Output]
[Task]
<Question Answering (QA)>
Things to do for healthy muscle growth

[Constraint]
<Length>
Less than 150 characters <Format>
Bullet point
<Numeric>
three bullet point
word protein at least 8 times
word muscle appears less than 10 times
<Content>
(Include) Ends with "Thank you"
(Exclude) Don't start with "Yes, I understand"

Example 2
...

[Caution]
1. If the output length is described qualitatively, follow these criteria:
If expressions like "briefly", "short", "concise", or "summarize in a few words" are present → (Short)"briefly", "short", "concise", "summarize in a few words"
If expressions like "detailed", "comprehensive", "in-depth", "elaborate", "thorough", or "explain in detail" are present → (Long)"detailed", "comprehensive", "in-depth", "elaborate", "thorough", "explain in detail"
2. If multiple tasks are requested and are related (e.g., both Question Answering and Summarization), each task should be classified separately:
   - Even if tasks are interconnected (e.g., answering a question and summarizing the answer), they should be treated as distinct tasks.
   - Each task's objective and constraints must be captured separately to ensure clarity and accuracy in classification.
3. Extractions related to the quantity of a specific target, excluding the length of the entire text, should be extracted to <Numeric>.
4. Your generation should be only [Output] which contains [Task] and [Constraint], notes or explanations are strictly prohibited.

Consider [Caution] and classify [Task] and [Constraint] in the below [Instruction].

[Output] must strictly follow the format like above examples. No other correction, step for explanation is allowed.
Generate only one final [Output] in the format of above examples.
[Instruction]
{instruction}

## C.2 FEEDBACK

Feedback prompt is zero-shot prompt which request correction of previous generation. It consists request for correction and caution about unnecessary additional explanations that often appear in creation statements.

> **Feedback**
>
> [Previous Generation]
> {previous_generation}
>
> [Feedback]
> {previous_feedback}
>
> [Instruction]
> {question}
> —
> Your job is to correct the given [Previous Generation] based on the [Feedback] to follow the [Instruction] and generate only one final corrected generation.
> Don't include any explanation or note. Just generate the corrected generation.

## C.3 STRUCTURE EVALUATION

We defined Structure Evaluation as a dichotomous problem, and gave it a score of 5 if there was no problem and 0 if there was a problem in structure of LLM generation. A total of 5 common and problematic examples are provided.

> **Structure Evaluation**
>
> You are a response quality evaluator. Your task is to determine whether a given AI-generated response is acceptable in terms of relevance, coherence, and content quality. You have to provide score and overall feedback for improvement of judgement in designated structure.
> —
> Evaluate each response as follows:
> Score : 5
> The response is coherent, well-structured, and grammatically correct.
> It maintains a clear and consistent topic throughout.
> It does not include irrelevant filler, excessive repetition, or multiple alternate responses.
> Brief or concise responses are acceptable as long as they maintain clarity and correctness.
> Score : 0
> The response contains list of meaningless words, incoherent, ungrammatical content.
> It provides multiple draft versions in a single output.
> —
> Examples
> —
> Example 1
> [Generated Response]
> Hormones are chemical messengers produced by glands or organs in our body, which play crucial roles in regulating various physiological processes such as growth, development,

metabolism, reproduction, and many others. To perform their functions effectively, hormones must be able to interact with cells throughout the body. However, for them to do so efficiently, they need to have certain characteristics that enable them to pass through the **cell membrane**, a thin layer surrounding every cell composed mainly of lipid molecules embedded with proteins. The ability of hormones to traverse this barrier depends on how well they dissolve in lipids - those being **lipid-soluble** will find it easier compared to water-soluble ones due primarily because the main component of a typical **cell membrane** is made up largely from these same kinds.

The reason why some types of hormones can easily penetrate into cells lies within their inherent properties related specifically towards solubility; meaning whether something dissolves better either inside fat-based substances like oils (hence called hydrophobic) versus aqueous mediums akin more closely resembling plain old H2O itself! Given its predominantly fatty nature then naturally speaking here too only similarly suited partners would readily make passage past said protective gateways known otherwise commonly referred simply enough under common language terms again: namely "the" singular very critical entity indeed dubbed still basically yet extremely aptly put — ''a **cell membrane"''. Consequently therefore finding themselves freely circulating everywhere allowing direct unhindered movement while absolutely necessitating higher concentrations elsewhere simultaneously continuing functioning optimally without breaking down excessively thus encompassing truly vital life maintaining regulatory controlling functionalities across diverse bodily domains reaches far-flung distances unobstructed reaching even crossing several layers all eventually converging upon final destination sites harmoniously integrating fully interlacing biological pathways systemically unlocking useful synergistic benefits overall biodiversity thriving beautifully exemplified perfectly say via Thyroid Hormone – one prime illustration fitting neatly right into aforementioned categorizations belonging firmly amongst ranks highly prized especially valued members recognized universally definitely qualifying squarely meeting necessary distinct stringent criteria thanks significantly partly really rather fundamentally owed uniquely intrinsic special fundamental molecular structural attributes rendering particularly exceedingly rich profoundly enabling nonchalantly traverses multiple respective barriers including notably most critically its undoubted capacity penetrating just about instantly straightaway whenever originating signals impinge forcefully triggering immediate localized powerful adaptations demanding unwavering swift vigorous unremitting responses anywhere seamlessly passing albeit always effortlessly gliding unfettered almost frictionlessly onward bound easily surmountable jumping over frequently encountered obstacles posed typically restrictive selectively filtering gates kind dynamically

[Judgment]
{"Score" : "0", "Overall Feedback" : "The context is overly verbose, repetitive, and lacks clear structure, making it hard to follow. It uses unnecessarily complex phrasing and deviates from the main point, which weakens the overall coherence and effectiveness of the explanation.Focus on clarity and conciseness. Remove repetitive explanations and use simpler, more direct language to communicate the main ideas. Keep the content organized and ensure each sentence builds logically on the previous one."} —

Example 2
...
—
[Generated Response]
{generated_response}

[Judgment]
—
Classify the above [Generated Response].
Generate only the [Judgment] in the designated format.
[Judgment] with no Score value and Overall Feedback value is prohibited.
End generation when [Judgment] is over.

## C.4  CONTENT EVALUATION

### C.4.1  TASK EVALUATION

Criteria for evaluating each task are set, and criteria for measuring sub-scores from 1 to 5 are described. Examples for each sub-score and corresponding feedback are described.

---

**Question Answering Task Evaluation**

Please rate whether the generated answer to the following question is semantically similar to the correct answer provided and provide overall feedback.. Instead of a binary (100 or 0 points) evaluation, use a 1 to 5 rating scale based on the degree of semantic similarity.

—

[Evaluation Criteria]
Each criterion is rated from 1 to 5, with a maximum total score of 5 points.
1. Semantic Similarity
   - Does the generated answer correctly convey the key information required to answer the question?
   - If the key information is correctly included, the answer should be considered correct, even if additional details differ from the correct answer.
   - Minor omissions of non-essential details should not lower the score significantly.
   - However, if critical information is missing or incorrect, the score should be reduced.

   Scoring Guide:
      - (1) Incorrect Semantic Similarity → The generated answer is unrelated or completely incorrect.
      - (2) Minimal Semantic Similarity → The answer is somewhat related but does not provide the correct key information.
      - (3) Somewhat Similar → The answer includes key information but is incomplete or contains inaccuracies.
      - (4) Mostly Similar → The key information is correct, but some secondary details differ or are missing.
      - (5) Perfect Match → The key information is correctly stated, and additional details (if any) are relevant and accurate.

—

[Evaluation Examples]

Example 1: Perfect Semantic Match

[Main goal]
chemical symbol for water

[Correct Answer]
$H_2O$

[Generated Answer]
$H_2O$

[Evaluation result]
{"Score" : "5", "Overall Feedback" : "The generated answer is fully correct and matches the expected response."}
—
Example 2
...

—
Consider [Evaluation Criteria] and refer to the [Main goal] to evaluate whether [Generated Answer] and [Correct Answer] are semantically similar.
Generate only one final [Evaluation result] in the format of above examples.
[Evaluation Result] generation with no Score value and Overall Feedback value is prohibited.
—
[Main goal]
{main_goal}

[Correct Answer]
{ground_truth}

[Generated Answer]
{generated_answer}

[Evaluation result]

---

### Summarization Task Evaluation

The following prompt evaluates how accurately the summary generated from the provided document or models answer contains key information and provide overall feedback. The summary is assessed based on accuracy, coverage of key information, and conciseness, with each category rated on a 1 to 5 scale.
—
[Evaluation Criteria]
1. Accuracy
   - Does the summary correctly represent the key information from the original text?
   - Ensure there is no misinformation, distortion of meaning, or hallucinated content.
   - Essential details should be accurately conveyed.

   Scoring Guide:
      - (1) Incorrect $\rightarrow$ Contains misinformation or significantly distorts meaning.
      - (2) Limited Accuracy $\rightarrow$ Some information is correct, but key details are missing or altered.
      - (3) Partially Accurate $\rightarrow$ Covers some essential aspects but omits or modifies important details.
      - (4) Mostly Accurate $\rightarrow$ Largely correct but may slightly misrepresent or leave out minor details.
      - (5) Fully Accurate $\rightarrow$ Faithfully represents all key information without errors.

2. Coverage of Key Information
   - Does the summary capture essential concepts such as people, events, places, and figures?    - Deduct points if critical elements are missing.

   Scoring Guide:
      - (1) Insufficient Coverage $\rightarrow$ Omits most essential concepts.
      - (2) Minimal Coverage $\rightarrow$ Includes a few key details but misses many critical aspects.
      - (3) Partial Coverage $\rightarrow$ Covers some essential elements but lacks completeness.
      - (4) Mostly Covered $\rightarrow$ Contains most key concepts but misses minor elements.
      - (5) Fully Covered $\rightarrow$ Captures all essential concepts from the original text.

3. Conciseness

- Is the summary clear and to the point without unnecessary details?
- Does it condense the content effectively while retaining meaning?

Scoring Guide:
- (1) Overly Redundant or Too Brief → Either too lengthy with unnecessary details or too short, omitting key information.
- (2) Somewhat Redundant or Overly Condensed → Contains excessive or missing details, affecting clarity.
- (3) Balanced but Slightly Off → Generally concise but could be refined for better clarity.
- (4) Well-Condensed → Summarized effectively with minimal redundancy.
- (5) Optimally Concise → Clear, concise, and well-balanced.

—

[Evaluation Examples]

Example 1: Highly Accurate Summary (Score: 5/5)
[Target]
Seoul, the capital of South Korea, serves as the nation's political, economic, and cultural hub. It is home to key government institutions, major corporations, and renowned cultural landmarks. The city boasts a dynamic economy, driven by thriving IT and financial industries, which position it as a global center for innovation and business. With its advanced infrastructure and vibrant cultural scene, Seoul continues to shape South Korea's growth and international influence.

[Main goal]
Summarize the economic importance of Seoul.

[Generated Answer]
Seoul is the capital of South Korea and a hub for politics, economy, and culture, with thriving IT and financial industries.
—
[Evaluation result]
{"Score" : "5", "Overall Feedback" : "The summary effectively conveys all key details while maintaining clarity and conciseness."}
—
Example 2
...
—
Consider [Evaluation Criteria] and refer to the [Main goal] to evaluate whether [Generated Answer] summarizes the [Target].
Generate only one final [Evaluation result] in the format of above examples.
[Target], [Main goal], [Generated Answer] or other explanations are not allowed in your generation.
[Evaluation Result] generation with no Score value and Overall Feedback value is prohibited. —
[Target]
{target}

[Main goal]
{main_goal}

[Generated Answer]
{generated_answer}

[Evaluation result]

**Generation Task Evaluation**

This prompt evaluates whether the generated response sufficiently includes the content requested by the user and provide overall feedback. The assessment focuses only on the presence of key information, ensuring that the generated text fulfills the primary intent of the request.

—

[Evaluation Criteria]

Content Inclusion (Score: 1 to 5)
The generated response is evaluated based on how well it covers the key content requested by the user.

- 5 → The response fully addresses all essential aspects of the request with relevant details.
- 4 → The response mostly includes the requested content but lacks minor details.
- 3 → The response covers part of the requested content but is missing important elements.
- 2 → The response includes only a small portion of the requested content and omits key information.
- 1 → The response does not address the request meaningfully or is entirely off-topic.

—

Evaluation Examples

Case 1: Fully Meets the Request (Score: 5/5)

[Main goal]
short proposal for a new research project that investigates how language evolves over time.

[Generated Response]
Title: Investigating Language Evolution in the Digital Age

Objective:
This research project aims to analyze how language changes over time, particularly in response to digital communication and social media.
Methodology:
We will conduct a corpus-based analysis of historical and modern texts, tracking linguistic shifts in vocabulary, grammar, and usage patterns. Additionally, we will use machine learning models to predict future language trends.
Expected Outcomes:
This study will provide insights into the factors influencing language evolution and contribute to linguistic theory and AI-driven language modeling.

[Evaluation result]
{"Score" : "5", "Overall Feedback" : "The response fully meets the request by providing a structured and relevant research proposal. No revisions are necessary."}

—

...

—

Use the [Evaluation Criteria] to evaluate whether the [Generated Response] sufficiently includes the requested content from the [Main goal].
Generate only one final [Evaluation result] which in the format of above examples.
[Evaluation Result] generation with no Score value and Overall Feedback value is prohibited.

—

```
[Main goal]
{main_goal}

[Generated Response]
{generated_answer}

[Evaluation result]
```

### C.4.2 CONSTRAINT EVALUATION

Evaluation criteria have been established for each constraint, with detailed scoring guidelines ranging from 1 to 5. Examples corresponding to each score level, along with the associated feedback, are also provided.

---

**Format Constraint Evaluation**

This prompt evaluates whether the generated response adheres to multiple specified format constraints and provide overall feedback. The assessment ensures that the response strictly follows structural, stylistic, or formatting requirements.

—

[Evaluation Criteria]

Format Constraint Adherence (Score: 1 to 5)

- 5 → The response strictly follows all format constraints with no deviations.
- 4 → The response mostly follows the format but has minor inconsistencies.
- 3 → The response partially follows the format but misses some key elements.
- 2 → The response loosely follows the format but requires major improvements.
- 1 → The response does not follow the format at all.

—

[Evaluation Examples]

Case 1: Strictly Follows All Constraints (Score: 5/5)

[Generated Response]
"Dear Hiring Manager,

I am excited to apply for the software engineer position at your esteemed company. I have a strong background in backend development and AI-driven applications.

My qualifications include:
- Proficiency in Python and Java
- Experience in cloud computing
- Strong problem-solving skills

I look forward to your response. Best regards, John Doe."

[Format Constraints]
Two paragraphs
Includes a bullet-pointed list
Ends with a polite closing

[Evaluation Result]
{"Score": "5", "Overall Feedback": "The response strictly follows all format constraints, including paragraph structure, bullet points, and a polite closing."}
—

Case 2
...
—

Use the [Evaluation Criteria] above to determine whether the [Generated Response] meets the [Format Constraints].
Only the provided [Format Constraints] is the target for evaluation, no other constraint must be evaluated.
[Evaluation Result] with no Score value or no Overall Feedback value is prohibited.
Generate only one final [Evaluation Result] which strictly follows the format of above examples.
—
[Generated Response]
{generated_answer}

[Format Constraints]
{format_constraints}

[Evaluation result]

---

## Numeric Constraint Evaluation

This prompt evaluates whether the generated response fulfills the numeric constraints specified by the user and provide overall feedback. The evaluation focuses on the correct count, occurrences, or specific quantities in the generation.
—
[Evaluation Criteria]
Numeric Constraint Adherence (Score: 1 to 5)
5 → The response strictly adheres to all numeric constraints, with no deviations or errors.
4 → The response mostly adheres to the numeric constraints but contains minor errors.
3 → The response partially meets the numeric constraints but misses some key elements.
2 → The response loosely follows the numeric constraints with significant errors.
1 → The response does not meet the numeric constraints at all.

Evaluation Examples

Case 1: Strictly Fulfills All Numeric Constraints (Score: 5/5)

[Generated Response]
Solar energy harnesses sunlight using solar panels, which convert sunlight into electricity.
Solar power is a renewable source of energy, providing an eco-friendly alternative to fossil fuels.
With advancements in technology, solar energy is becoming increasingly efficient and affordable for both homes and businesses.

[Numeric Constraints]
Exactly 3 Bullet points
Word 'solar' appears more than 2 times
Word 'energy' appears exactly 4 times

[Evaluation Result]

{"Score": "5", "Overall Feedback": "The response strictly meets all the numeric constraints. It has exactly 3 bullet points, the word 'solar' appears more than 2 times, and 'energy' appears exactly 4 times."}

—

Case 2
...
—

Use the [Evaluation Criteria] above to determine whether the [Generated Response] meets the [Numeric Constraints].
Only the provided [Numeric Constraints] is the target for evaluation, no other constraint must be evaluated.
[Evaluation Result] generation with no Score value and Overall Feedback value is prohibited.
Generate only one final [Evaluation Result] which strictly follows the format of above examples.
—

[Generated Response]
{generated_answer}

[Numeric Constraints]
{numeric_constraints}

[Evaluation result]

---

### Length Constraint Evaluation

This prompt evaluates whether the given number, which is the word count of generated response of AI assistant, meets the range of specific constraints and provide overall feedback.
—

[Evaluation Criteria]

Number Constraints Adherence (Score: 0 or 5)

- 5 → All numbers satisfies the constraints.
- 0 → Some numbers doesn't meet the constraints.

—

Evaluation Examples

Case 1: All numbers meet the constraints (Score: 5/5)

[Number]
424 words

[Constraints]
More than 300 words

[Evaluation Result]
{"Score" : "5", "Overall Feedback" : "The generated response satisfies the constraints More than 300 words with 424 words."}
—

Case 2:
...
—

[Number]
{number}

[Constraints]
{length_constraint}

[Evaluation result]

–

Use the [Evaluation Criteria] above to determine whether the above [Number] is in the range of the [Constraints].
[Evaluation Result] generation with no Score value and Overall Feedback value is prohibited.
Strictly follow the format of [Evaluation result] like above examples.
Generate only one final [Evaluation Result] which strictly follows the format of above examples.

---

## Content Constraint Evaluation

This prompt evaluates whether the generated response adheres to the given content constraints, which specify words, phrases, or sentence structures that must or must not be included and provide overall feedback.

—

Evaluation Criteria and Scoring Method

Each response is rated on a scale from 1 to 5, based on how strictly it follows the given constraints:

1. (Score: 5) Fully adheres to provided content constraints.
2. (Score: 4) Minor deviation (e.g., a partial mismatch in phrase placement or slight variation in required content).
3. (Score: 3) Moderate deviation (e.g., one constraint completely ignored but others followed).
4. (Score: 2) Significant deviation (e.g., multiple constraints ignored).
5. (Score: 1) Completely fails to meet provided content constraints.

—

Evaluation Examples

Case 1: Fully Adheres to Content Constraints (Score: 5/5)

[Generated Response]
"Hard work and perseverance always lead to success. No matter how difficult the journey

is, every step forward is progress. Keep pushing ahead and never give up. Good job."

[Content Constraints]
- Ends with "Good job"
- Does not contain "failure"
- Contains "success" at least once

[Evaluation Result]

{"Score": "5", "Overall Feedback": "The response fully meets the content constraints. It ends with 'Good job,' includes the word 'success,' and does not contain 'failure'."}

—
Case 2
...
—

Use the [Evaluation Criteria] and [Content Constraints] to evaluate whether [Generated Answer] meets the [Content Constraints].
Only the provided [Content Constraints] is the target for evaluation, no other constraint must be evaluated.
[Evaluation Result] generation with no Score value and Overall Feedback value is prohibited.
Generate only one final [Evaluation Result] which strictly follows the format of above examples.
—

[Generated Answer]
{generated_answer}

[Content Constraints]
{content_constraints}

[Evaluation result]

