# OpenReview forum: "Efficient Self-Review Framework for Enhancing Instruction Following Capability of LLM"
_ICLR.cc/2026/Conference — Submitted to ICLR 2026_

### Official Review · Reviewer_w7Fc · 2025-10-31

**Soundness:** 2
**Presentation:** 2
**Contribution:** 2
**Rating:** 4
**Confidence:** 3

**Summary:**

The paper proposes Re5, a structured self-revision framework that decomposes user instructions into task and constraint components, conducts structure-aware self-evaluation, and performs selective constraint-specific correction. Re5 aims to improve instruction following of open LLMs (LLaMA 3.3-70B) efficiently, without over-reliance on high-cost teacher models. Experiments show performance on IFEval and Multi-IF comparable to GPT-4o-mini–supervised models.

**Strengths:**

1. The paper aims to address a meaningful issue: instruction-following improvements often trade off output quality or require costly teacher supervision.

**Weaknesses:**

1. While the combination of structured constraint evaluation + selective correction is useful, the paper’s conceptual contribution is incremental relative to existing works, such as Self-Refine [1], DeCRIM [2]. The authors should explicitly delineate how Re5 differs in algorithmic mechanism and also compare to these methods in experiments.

2. Efficiency is claimed but not quantified beyond smaller dataset size. There is no compute-time or generation-token comparison provided.

3. The OQA heavily depends on GPT-4o-mini as judge, which may favor GPT-style generations and bias results. A small scale human evaluation or cross-judge validation (e.g., Claude or Mixtral) would be more convincing. Augmenting Natural Questions and XL-Sum with GPT-4o-mini generated constraints raises data contamination and fairness issues since the teacher model influences both data and evaluation.


[1] https://arxiv.org/abs/2303.17651
[2] https://arxiv.org/abs/2410.06458

**Questions:**

1. Could the “structured feedback extraction” step be replaced by automated parsing (e.g., JSON validation)?
2. Have you tested Re5 on smaller open models (e.g., 7B) to confirm scalability claims?
3. Please add white space between the content and citation. Line 168, please cite the paper published version instead of anonymous.

---

### Official Review · Reviewer_zCK4 · 2025-10-31

**Soundness:** 3
**Presentation:** 3
**Contribution:** 3
**Rating:** 6
**Confidence:** 3

**Summary:**

This paper propose Re5, a framework that leverage self-evaluation and revision capability of LLMs to refine their responses. By decomposing instructions, perform structure-aware evaluations and generate feedbacks for revision, Re5 improves both the instruction adherence and the overall quality of the responses. The refined responses are then used in alignment fine-tuning to further enhance the instruction following capability of LLMs. Experiments demonstrate that Re5 helps enhance model's instruction following capability with less data compared with other alignment-tuning methods.

**Strengths:**

1. The framework is concise and easy to construct, providing an applicable approach for LLMs' self-evaluation and self-enhancement.
2. The ablation experiment shows the significance of each process in Re5 framework, demonstrating the validity of the framework design.

**Weaknesses:**

1. The main experiment in Table 1 only use a single base model Llama-3.3-70B-Instruct, which cannot effectively demonstrate the generality of the method across various models. It is necessary to supplement experiments on models of different series and scales.
2. Current results cannot show the advancement of Re5 compared with other self-review methods that improve the response quality. The main experiment also lacks comparison results with these methods, such as self-review through prompting and other approaches.
3. The citation format are mistakenly used in multiple parts in the paper. When the authors or the publication are not included in the sentence, the citation should be in parenthesis (e.g. in line 45-46, it should be "a means to improve instruction-following capabilities (Pan et al. 2024)" instead of "a means to improve instruction-following capabilitiesPan et al. (2024)").

**Questions:**

See weaknesses.

---

### Official Review · Reviewer_HpU1 · 2025-11-02

**Soundness:** 2
**Presentation:** 2
**Contribution:** 2
**Rating:** 4
**Confidence:** 2

**Summary:**

The paper introduces Re5, a self-evaluation and revision framework designed to improve the instruction following ability of large language models while maintaining output quality and efficiency.
Rather than depending on high-cost external evaluators (e.g., GPT-4 level judges) or multiple full regeneration cycles, Re5 decomposes user instructions into task and constraint components, such as format, numeric, length, and so on.

**Strengths:**

Clear motivation: recognizes the dual challenge of instruction adherence vs response quality, a gap in many self-revision works.

**Weaknesses:**

1. All experiments use a single base model (LLaMA 3.3 70B). Cross model or smaller model generalization is only discussed as future work.

2. No human rated comparison of readability, coherence, or factual faithfulness, important for verifying “quality preservation.”

3. Though multiple public corpora are adapted (NQ, XL-Sum, Tulu 3), details on GPT-4o-mini augmentation prompts and constraint distributions are brief, leaving potential data leakage or bias unexamined.

**Questions:**

Please refer to the weakness part.

---

### Meta-Review · Area_Chair_iwNS · 2026-01-11

**Summary:**

The paper argues that LLMs often partially follow complex instructions—especially when there are multiple constraints (format, length, counts, must-include/must-avoid phrases, etc.). Prior “revise until compliant” approaches can be expensive and can damage output quality. Re5 proposed an engineering-heavy self-review framework for this, that combines: (i) constraint decomposition, (ii) structural gating to prevent degenerate revisions, (iii) constraint-wise scoring with structured feedback, and (iv) selective correction rather than full regeneration.

However, reviewers converge on several missing pillars: generality (only LLaMA 3.3 70B), efficiency accounting (no tokens/time cost), stronger baseline comparisons (Self-Refine/DeCRIM-style), and quality validation beyond GPT-4o-mini judging, especially given GPT-4o-mini’s role in data augmentation and judging.

**Reviewer Concerns:**

No rebuttal in this submission

Still outstanding:

1. Cross-model / small-model generalization
2. Compute/time/token efficiency comparison
3. Head-to-head comparisons vs established self-revision methods
4. Human eval / cross-judge validation and contamination analysis for GPT-4o-mini augmentation + evaluation

**Reviewer Scores:**

No rebuttal in this submission

---

### Decision · Program_Chairs · 2026-01-26

Reject